# Study of the Optical Features of Tb³⁺:CaYAlO₄ and Tb³⁺/Pr³⁺:CaYAlO₄ Crystals for Visible Laser Applications

**Yeqing Wang [1], Jian Cheng [1], Zhiyuan Wang [1], Yujing Gong [1], Chaoyang Tu [2,3], Jianhui Huang [2,3], Yijian Sun [2,3,\*] and Yi Yu [4,\*]**

[1] Department of Applied Physics, East China Jiaotong University, Nanchang 330013, China
[2] Jiangxi Provincial Key Laboratory of Functional Molecular Materials Chemistry, School of Chemistry and Chemical Engineering, Jiangxi University of Science and Technology, Ganzhou 341000, China
[3] National Rare Earth Functional Material Innovation Center, Ganzhou 341000, China
[4] School of Physics and Electronics Information, Gannan Normal University, Ganzhou 341000, China
[\*] Correspondence: sunyijian@jxust.edu.cn (Y.S.); yuyignnu@163.com (Y.Y.)

**Abstract:** Single crystals of Tb³⁺ single-doped and Tb³⁺/Pr³⁺ co-doped CaYAlO₄ were produced by the Czochralski method. The room-temperature polarized absorption spectra, emission spectra, and decay curves were recorded and analyzed in detail. The absorption cross-section around 487 nm was found to be $1.53 \times 10^{-22}$ cm² for the π polarization in the Tb³⁺:CaYAlO₄ crystal and increased to $5.23 \times 10^{-22}$ cm² in the Tb³⁺/Pr³⁺:CaYAlO₄ crystal. The spectroscopic parameters were calculated through the Judd–Ofelt theory. For the Tb³⁺:CaYAlO₄ crystal, the emission bands of green light at 546 nm and yellow light at 587 nm had fluorescence branching ratios of 64.7% and 6.65% with cross-sections of $8.82 \times 10^{-22}$ cm² (σ-polarization) and $0.44 \times 10^{-22}$ cm² (π-polarization), respectively. The decay lifetimes of ⁵D₄ multiplets were measured to be 1.41 ms and 1.1 ms for Tb³⁺:CaYAlO₄ and Tb³⁺/Pr³⁺:CaYAlO₄ crystals, respectively. The energy transfer mechanisms of Tb³⁺ and Pr³⁺ and their emission spectral intensities at different temperatures were analyzed. As the temperature increased, the luminescence intensity of the Tb³⁺:CaYAlO₄ and Tb³⁺/Pr³⁺:CaYAlO₄ crystals decreased almost linearly with the CIE coordinate variation, from (0.370, 0.621) to (0.343, 0.636) and from (0.345, 0.638) to (0.246, 0.698), respectively. The results indicate the potential of Tb³⁺:CaYAlO₄ and Tb³⁺/Pr³⁺:CaYAlO₄ crystals as visible laser materials with a wide temperature range.

**Keywords:** Tb³⁺:CaYAlO₄; Tb³⁺/Pr³⁺:CaYAlO₄; spectroscopic characteristics; visible emission

## 1. Introduction

Solid-state lasers in the visible band have a variety of applications, including biomedical instrumentation, visual displays, and remote sensing [1–3]. There are several reports about the operation of visible solid-state lasers. One example is the 589 nm laser produced by 1064 and 1319 nm lasers through sum-frequency mixing from a Nd:YAG crystal [4,5]. Second-harmonic generation (SHG) or sum-frequency generation takes place in lithium triborate crystals, producing visible outputs at any of the following three wavelengths: 537 nm, 546 nm, and 556 nm [6]. Moreover, the appropriate configuration of a He-Ne laser can emit laser beams at 594 nm and 612 nm. Although nonlinear optical technology is used in practice, the adoption of these methods may lead to a complex and expensive optical system, complicated operation, and poor beam quality, restricting their further development and application. Thus, it is of great scientific significance to discover other routes to produce visible lasers. Today, thanks to the rapid development of laser diodes (LDs) in the blue region [7–9], the output of green and yellow lasers has been achieved by LD-pumped visible laser gain media. For example, in 2020, 622 nm, 662 nm, and 747 nm lasers were produced with a YAlO₃:Pr³⁺ crystal pumped by a 488 nm semiconductor laser [10]. Chen et al. reported a Tb³⁺:LiYF₄ laser with maximum output power

of 1.17 W (@544 nm) and 0.5 W (@587 nm) [11,12]. This method avoids the complex non-linear frequency conversion and has the properties of compact structure, high stability, good beam quality, etc., playing an increasingly vital role in visible laser techniques.

As is widely known, based on the energy level of $Tb^{3+}$, the emission bands around 546 nm and 578 nm are located in the green and yellow ranges, respectively, corresponding to the $^5D_4 \rightarrow {}^7F_J$ (5, 4) transition [13]. According to previous investigations, $Tb^{3+}$ was introduced to some fluoride host materials, such as $CaF_2$, $CdF_2$, and $LiYF_4$ [14–16], which commonly suffer much energy waste and poor physical and chemical properties. As an alternative choice, oxides have higher mechanical strength and better chemical stability for lasing operations. The structure of $CaYAlO_4$ (abbreviated as CYA) crystal is highly disordered, and its lattice parameters are a = b = 3.6451 Å and c = 11.8743 Å [17].

However, the transition of $Tb^{3+}$:$^7F_6 \rightarrow {}^5D_4$ is a spin-forbidden process, resulting in a relatively small absorption cross-section around 487 nm, at a magnitude of $10^{-22}$ $cm^2$ [14]. Higher $Tb^{3+}$ concentrations or co-doping with rare-earth ions are the commonly used methods to overcome its weak absorption in practical applications. The energy level of $Pr^{3+}$:$^3P_0$ is very similar to that of $Tb^{3+}$:$^5D_4$ (as shown in Figure 1); the energy migration between these two states may be able help improve the small absorption cross-section of $Tb^{3+}$.

In our work, a $Tb^{3+}$ single-doped CYA crystal was produced via the Czochralski method. The spectral characteristics of the crystal were measured. In order to explore the effect of $Pr^{3+}$ on the low absorption cross-section of $Tb^{3+}$ around 487 nm, a $Tb^{3+}$/$Pr^{3+}$ co-doped CYA crystal was grown through the same growth technique. The energy migration route between $Tb^{3+}$ and $Pr^{3+}$, as along with the effect of temperature on the fluorescence emission, was displayed and studied for the exploration of their laser prospects.

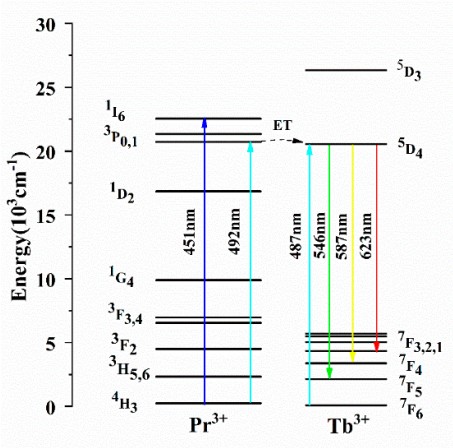

**Figure 1.** The energy level diagrams of $Pr^{3+}$ and $Tb^{3+}$ in CYA crystal.

## 2. Materials and Methods

Single crystals of 10 at.% $Tb^{3+}$ single-doped and 10% $Tb^{3+}$/0.6 at.% $Pr^{3+}$ co-doped CYA were produced by the Czochralski method. The Tb$^{3+}$:CYA and Tb$^{3+}$/Pr$^{3+}$:CYA polycrystalline materials with formulae of $CaY_{0.9}Tb_{0.1}AlO_4$ and $CaY_{0.84}Pr_{0.06}Tb_{0.1}AlO_4$, respectively, were prepared using high-temperature solid-state technology. The original materials used were $CaCO_3$ (AR grade, Sinopharm, Beijing, China), $Al_2O_3$ (AR grade, Sinopharm, Beijing, China), $Y_2O_3$ (99.99%, Changchun, China), $Tb_4O_7$ (99.99%, Changchun, China), and $Pr_6O_{11}$ (99.99%, Changchun, China) powders. The specific experimental process for the crystal growth was as described in [18]. Dark green Tb$^{3+}$:CYA and Tb$^{3+}$/Pr$^{3+}$:CYA crystals with almost the same size of $\Phi18 \times 18 \times 25$ $mm^3$ were obtained, as shown in Figure 2. The as-grown crystals were reheated in a flowing $N_2$(95%)–$H_2$(5%) mixture atmosphere at 1000 °C for 48 h to remove their intrinsic color center. The concentrations of $Tb^{3+}$ and $Tb^{3+}$/$Pr^{3+}$ in the single- and co-doped as-grown crystals were determined to be

13.87 at.% ($1.87 \times 10^{21}$ cm$^{-3}$) and 13.71 at.% ($1.75 \times 10^{21}$ cm$^{-3}$)/0.38 at.% ($0.477 \times 10^{20}$ cm$^{-3}$), respectively, by the inductively coupled plasma atomic emission spectrometry method (ICP-AES).

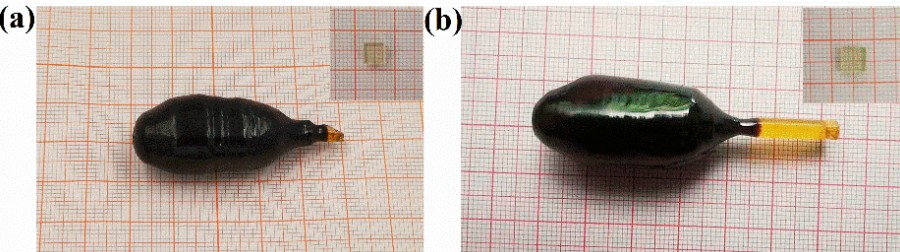

**Figure 2.** The as-grown (**a**) Tb$^{3+}$:CYA and (**b**) Tb$^{3+}$/Pr$^{3+}$:CYA crystals; the inserts are samples for spectral measurement with dimensions of $5 \times 5 \times 2$ mm$^3$.

The XRD patterns of the two obtained crystals were studied by X-ray diffraction (Miniflex600, Rigaku, Japan). Samples with dimensions of $5 \times 5 \times 2$ mm$^3$ were cut from the annealed crystals and optically polished for spectral measurement. The room-temperature polarized absorption spectra in the range of 300 nm–2500 nm were recorded using a PerkinElmer UV-VIS-NIR Spectrometer (Lambda-900, PerkinElmer, Ma, American). The fluorescence spectra and the appropriate lifetime decay curves were recorded at room temperature using FLS920 and FSP980 (Edinburg, England) spectro-photometers, respectively. The measurement conditions for the spectra remained the same for both samples to enable data comparisons.

## 3. Results and Discussion

### 3.1. X-ray Diffraction Analysis

The X-ray diffraction patterns of the Tb$^{3+}$:CYA and Tb$^{3+}$/Pr$^{3+}$:CYA crystals were studied and are shown in Figure 3. The diffraction peaks of the Tb$^{3+}$:CYA and Tb$^{3+}$/Pr$^{3+}$:CYA crystals were in good agreement with those of pure CYA crystal (PDF#24-0221). No other impurity peaks were detected, indicating that the as-grown crystals had a K$_2$NiF$_4$-type structure with an $I4/mmm$ space group.

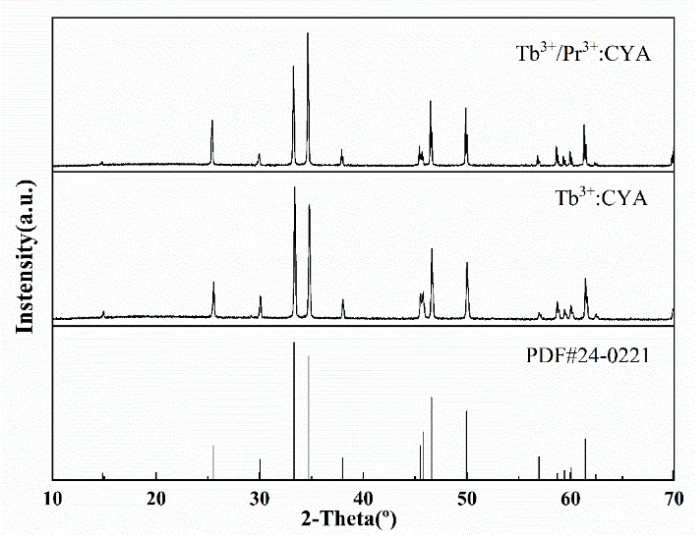

**Figure 3.** The XRD patterns of the Tb$^{3+}$:CYA and Tb$^{3+}$/Pr$^{3+}$:CYA crystals.

### 3.2. Absorption Spectra

The room-temperature polarized absorption spectra of the $Tb^{3+}$ single-doped and $Tb^{3+}/Pr^{3+}$ co-doped CYA crystals are shown in Figure 4. There are eight distinct absorption bands related to transitions from the ground multiplet $^7F_6$ to the excited multiplets of the $Tb^{3+}$, which are also indicated in Figure 4, as are the transitions of the $Pr^{3+}$ absorption band from its ground state $^3H_4$ to its excited state. In Figure 4, one can see that the weak absorption peaks of $Tb^{3+}$ are located at 320 nm, 340 nm, 351 nm, 370 nm, 380 nm, and 487 nm, corresponding to the $^7F_6 \rightarrow {}^5H_7 + {}^5D_{0,1}$, $^5L_6 + {}^5L_{7,8} + {}^5G_3$, $^5L_9 + {}^5G_4 + {}^5D_2$, $^5L_{10}$, $^5D_3 + {}^5G_6$, and $^5D_4$ transitions in the visible range, respectively. We can see two strong absorption peaks located around 1984 nm and 2293 nm in the near-infrared region, corresponding to transitions from $^7F_6$ to the higher multiplets $^7F_J$ (J = 0,1,2,3), respectively. In those absorption bands, the weak peak around 487 nm in the $^7F_6 \rightarrow {}^5D_4$ transition is consistent with commercial semiconductor lasers, which are commonly used as the pump source of $Tb^{3+}$ lasers. The $\pi$ and $\sigma$ polarization absorption cross-sections of $Tb^{3+}$:CYA at 487 nm were $1.53 \times 10^{-22}$ cm² and $1.55 \times 10^{-22}$ cm², which are smaller than those of $Tb^{3+}$:YAlO₃ ($3.0 \times 10^{-22}$ cm² @ 489 nm) but much larger than the value of $Tb^{3+}$:CaF₂ ($0.6 \times 10^{-22}$ cm² @ 485 nm) [14,19]. The $\pi$ and $\sigma$ polarization absorption cross-sections of $Tb^{3+}/Pr^{3+}$:CYA at 492 nm and 489 nm were $5.23 \times 10^{-22}$ cm² and $4.04 \times 10^{-22}$ cm², respectively, which are much larger than that of $Tb^{3+}$:CYA. The full widths at half-maximum (FWHMs) of the $Tb^{3+}$:CYA crystal around 487 nm were measured to be 9.39 nm and 8.93 nm for $\sigma$ and $\pi$ polarization, respectively, which are larger than the values for $Sr_3Tb_2(BO_3)_4$ (8.5 nm at 486 nm) and $Tb^{3+}$:YAlO₃ (3.64 nm at 486 nm) [13,20]. The absorption cross-sections were strengthened, meaning that the co-doped $Pr^{3+}$ could be used to solve the problem of the weak absorption cross-section of the $^7F_6 \rightarrow {}^5D_4$ transition in $Tb^{3+}$.

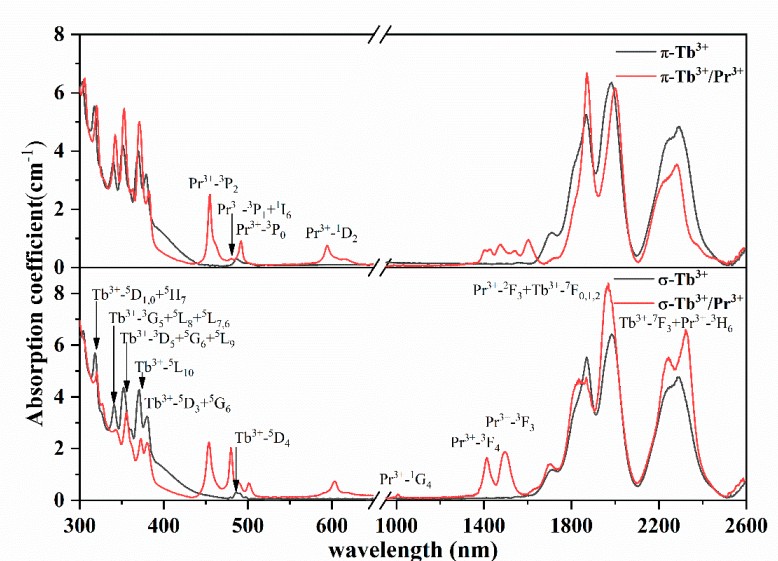

**Figure 4.** The room-temperature polarized absorption spectra of the $Tb^{3+}$:CYA and $Tb^{3+}/Pr^{3+}$:CYA crystals.

### 3.3. Judd–Ofelt Analysis

The spectral characteristics of the $Tb^{3+}$:CYA and $Tb^{3+}/Pr^{3+}$:CYA crystals were analyzed by the Judd–Ofelt (J–O) theory. The calculation process of the J–O theory is similar to that described in Ref. [20]. The mean wavelength ($\bar{\lambda}$) and the experimental and calculated line strengths for the $Tb^{3+}$:CYA and $Tb^{3+}/Pr^{3+}$:CYA crystals in both polarizations are listed in Tables 1 and 2, respectively. In Table 3, the calculated J–O intensity parameters of $Tb^{3+}$ in CYA and other crystals are listed. On account of the polarized absorption, the effective J–O intensity parameters can be obtained by $\Omega_{eff} = (\Omega_\pi + 2\Omega_\sigma)/3$. According to some previous works, $\Omega_2$ is a covalency-dependent parameter, while $\Omega_4$ and $\Omega_6$ are

structure-dependent ones, and the former depends on covalent bonding between coordination ions and rare-earth ions [21,22]. The $\Omega_{eff,2}$ of $Tb^{3+}$ in the CYA crystal was much greater than that in $CaF_2$ and YAG, showing that a higher $\Omega_{eff,2}$ value means a higher covalency of the metal–ligand bond, along with low symmetry of the coordination structure around $Tb^{3+}$. The value of $\Omega_{eff,4}/\Omega_{eff,6}$ was 1.61 and 1.94 in the $Tb^{3+}$:CYA and $Tb^{3+}/Pr^{3+}$:CYA crystals, respectively, which are higher than the values in $LiYF_4$, YAG, and CGA, but smaller than that in YAP.

**Table 1.** Mean wavelength $\bar{\lambda}$ and experimental and calculated absorption line strengths of ED transitions of the $Tb^{3+}$:CYA crystal.

| Transitions | π-Polarization, S($10^{-20}$ cm²) | | | σ-Polarization, S($10^{-20}$ cm²) | | |
|---|---|---|---|---|---|---|
| $^6F_{7\rightarrow}$ | $\bar{\lambda}(nm)$ | $S_{exp}^{ED}$ | $S_{cal}^{ED}$ | $\bar{\lambda}(nm)$ | $S_{exp}^{ED}$ | $S_{cal}^{ED}$ |
| $^5H_7 + ^5D_{0,1}$ | 320 | 0.046 | 0.042 | 320 | 0.047 | 0.05 |
| $^5L_6 + ^5L_{7,8} + ^5G_3$ | 340 | 0.058 | 0.056 | 341 | 0.054 | 0.063 |
| $^5L_9 + ^5G_4 + ^5D_2$ | 351 | 0.085 | 0.082 | 352 | 0.106 | 0.092 |
| $^5L_{10}$ | 370 | 0.084 | 0.083 | 371 | 0.092 | 0.091 |
| $^5D_3 + ^5G_6$ | 380 | 0.037 | 0.028 | 381 | 0.029 | 0.038 |
| $^5D_4$ | 487 | 0.0077 | 0.015 | 487 | 0.022 | 0.0081 |
| $^7F_{0,1,2}$ | 1984 | 1.506 | 2.37 | 1984 | 2.339 | 1.61 |
| $^7F_3$ | 2293 | 1.175 | 1.347 | 2290 | 1.348 | 1.162 |

**Table 2.** Mean wavelength $\bar{\lambda}$ and experimental and calculated absorption line strengths of ED transitions of $Tb^{3+}$ in the $Tb^{3+}/Pr^{3+}$:CYA crystal.

| Transitions | π-Polarization, S($10^{-20}$ cm²) | | | σ-Polarization, S($10^{-20}$ cm²) | | |
|---|---|---|---|---|---|---|
| $^6F_{7\rightarrow}$ | $\bar{\lambda}(nm)$ | $S_{exp}^{ED}$ | $S_{cal}^{ED}$ | $\bar{\lambda}(nm)$ | $S_{exp}^{ED}$ | $S_{cal}^{ED}$ |
| $^5H_7 + ^5D_{0,1}$ | 320 | 0.051 | 0.061 | 320 | 0.036 | 0.048 |
| $^5L_6 + ^5L_{7,8} + ^5G_3$ | 340 | 0.096 | 0.092 | 341 | 0.024 | 0.032 |
| $^5L_9 + ^5G_4 + ^5D_2$ | 351 | 0.16 | 0.129 | 352 | 0.055 | 0.049 |
| $^5L_{10}$ | 370 | 0.151 | 0.171 | 371 | 0.085 | 0.046 |
| $^5D_3 + ^5G_6$ | 380 | 0.046 | 0.0033 | 381 | 0.018 | 0.025 |
| $^5D_4$ | 487 | 0.029 | 0.012 | 487 | 0.0052 | 0.0051 |
| $^7F_{0,1,2}$ | 1984 | 2.292 | 2.081 | 1984 | 2.776 | 0.869 |
| $^7F_3$ | 2293 | 0.962 | 0.962 | 2290 | 1.807 | 0.859 |

**Table 3.** J–O intensity parameters of different crystals doped with $Tb^{3+}$.

| Crystal | | $\Omega_2(10^{-20}$ cm²) | $\Omega_4(10^{-20}$ cm²) | $\Omega_6(10^{-20}$ cm²) | $\Omega_4/\Omega_6$ | Reference |
|---|---|---|---|---|---|---|
| $Tb^{3+}$:LiYF$_4$ | | 28.30 | 1.65 | 2.15 | 0.77 | [16] |
| $Tb^{3+}$:KYb(WO$_4$)$_2$ | | 1.91 | 2.41 | 4.91 | 0.49 | [23] |
| $Tb^{3+}$:CaF$_2$ | | 1.71 | 2.65 | 2.25 | 1.18 | [14] |
| $Tb^{3+}$:YAG | | 2.75 | 0.12 | 3.37 | 0.03 | [24] |
| $Tb^{3+}$:YAP | | 3.49 | 5.87 | 2.55 | 2.30 | [19] |
| $Tb^{3+}$:CYA | $\Omega_\pi$ | 3.79 | 2.58 | 1.4 | | |
| | $\Omega_\sigma$ | 4.25 | 2.31 | 1.51 | | |
| | $\Omega_{eff}$ | 4.1 | 2.4 | 1.47 | 1.63 | This work |
| $Tb^{3+}/Pr^{3+}$:CYA | $\Omega_\pi$ | 4.42 | 1.17 | 1.79 | | |
| | $\Omega_\sigma$ | 3.98 | 3.19 | 1.05 | | |
| | $\Omega_{eff}$ | 4.13 | 2.52 | 1.30 | 1.94 | |

The ED spontaneous transition rate ($A^{ED}$) was calculated on the basis of the obtained J–O parameters. The mean spontaneous transition rate (A) was obtained by $A = (A_\pi + 2A_\sigma)/3$ with $A = A_q^{ED} + A_q^{MD}$. Then, the fluorescence branching ratio (β) and radiation lifetime ($\tau_{rad}$) were assessed and tabulated, as shown in Tables 4 and 5, respectively, indicating that the transition $^5D_4\rightarrow^7F_5$ of $Tb^{3+}$ had the greatest β in both $Tb^{3+}$- and $Tb^{3+}/Pr^{3+}$-doped CYA crystals, with values of 64.7% and 64.8%, respectively. The $\tau_{rad}$ for the $^5D_4$ multiplets of the $Tb^{3+}$- and $Tb^{3+}/Pr^{3+}$-doped CYA crystals was calculated to be 1.805 ms and 1.86 ms, respectively—higher than the 1.7 ms recorded for $Tb^{3+}$:YAP [19]. Compared with the $Tb^{3+}$:CYA, the value of $\Omega_2$ was slightly larger in $Tb^{3+}/Pr^{3+}$:CYA, indicating a more disordered local symmetry of $Tb^{3+}$ in the co-doped crystal. This result was similar to that reported for a $Tb^{3+}/Pr^{3+}$:$CaF_2$ crystal by Liu [14].

**Table 4.** Spontaneous transition rates (A), fluorescence branching ratios (β), and radiation lifetime ($\tau_{rad}$) of the $Tb^{3+}$:CYA crystal.

| Transition | $A_\pi^{ED}$ (S⁻¹) | $A_\pi^{MD}$ (S⁻¹) | $A_\sigma^{ED}$ (S⁻¹) | $A_\sigma^{MD}$ (S⁻¹) | A (S⁻¹) | β(%) | $\tau_r$ (ms) |
|---|---|---|---|---|---|---|---|
| $^5D_4\rightarrow$ | | | | | | | 1.805 |
| $^7F_0$ | 16.907 | 0 | 15.166 | 0 | 15.746 | 2.84 | |
| $^7F_1$ | 10.661 | 0 | 9.563 | 0 | 9.929 | 1.79 | |
| $^7F_2$ | 14.184 | 0 | 15.192 | 0 | 14.856 | 2.68 | |
| $^7F_3$ | 43.888 | 0.288 | 47.899 | 0.286 | 46.848 | 8.45 | |
| $^7F_4$ | 36.483 | 0.215 | 35.898 | 0.215 | 36.308 | 6.55 | |
| $^7F_5$ | 331.998 | 2.631 | 368.493 | 2.631 | 358.959 | 64.7 | |
| $^7F_6$ | 69.889 | 0.907 | 70.83 | 0.907 | 71.42 | 12.8 | |

**Table 5.** Spontaneous transition rates (A), fluorescence branching ratios (β), and radiation lifetime ($\tau_{rad}$) of the $Tb^{3+}/Pr^{3+}$:CYA crystal.

| Transition | $A_\pi^{ED}$ (S⁻¹) | $A_\pi^{MD}$ (S⁻¹) | $A_\sigma^{ED}$ (S⁻¹) | $A_\sigma^{MD}$ (S⁻¹) | A (S⁻¹) | β(%) | $\tau_r$ (ms) |
|---|---|---|---|---|---|---|---|
| $^5D_4\rightarrow$ | | | | | | | 1.86 |
| $^7F_0$ | 2.756 | 0 | 14.634 | 0 | 10.67 | 2.95 | |
| $^7F_1$ | 1.783 | 0 | 9.23 | 0 | 6.75 | 1.86 | |
| $^7F_2$ | 15.196 | 0 | 8.777 | 0 | 10.92 | 2.71 | |
| $^7F_3$ | 55.856 | 0.288 | 25.42 | 0.277 | 35.85 | 8.48 | |
| $^7F_4$ | 27.734 | 0.215 | 26.317 | 0.207 | 27 | 6.64 | |
| $^7F_5$ | 429.988 | 2.631 | 182.682 | 2.54 | 270.35 | 64.8 | |
| $^7F_6$ | 62.88 | 0.907 | 47.885 | 0.878 | 53.77 | 12.74 | |

*3.4. Fluorescence Spectra*

The polarized fluorescence spectra of the $Tb^{3+}$:CYA and $Tb^{3+}/Pr^{3+}$:CYA crystals under the excitation of 487 nm and 492 nm, respectively, were recorded in the range of 500–725 nm, as shown in Figure 5. According to the energy level structure of $Tb^{3+}$, the visual-range emission bands located around 546 nm, 587 nm, 623 nm, 648 nm, 673 nm, and 683 nm correspond to the transitions of $^5D_4\rightarrow^7F_J$ (J = 5, 4, 3, 2, 1, 0), respectively, as indicated in Figure 5.

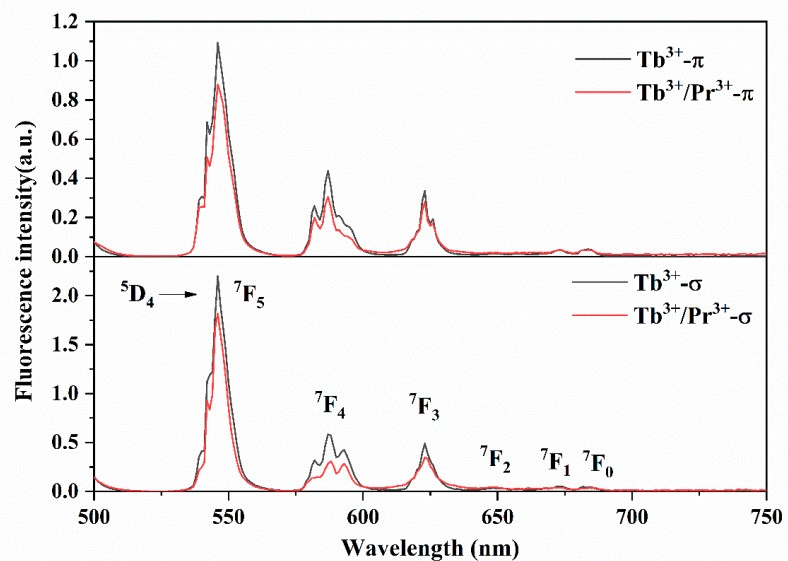

**Figure 5.** Room-temperature polarized fluorescence spectra of $Tb^{3+}$:CYA ($\lambda_{ex}$ = 487 nm) and $Tb^{3+}/Pr^{3+}$:CYA ($\lambda_{ex}$ = 492 nm) crystals in the 500–750 nm range.

As shown in Figure 5, the emission band shape of $Tb^{3+}/Pr^{3+}$:CYA was highly consistent with that of $Tb^{3+}$:CYA, due to the substantial coincidence of the fluorescence emission peaks of $Tb^{3+}(^5D_4\rightarrow)$ and $Pr^{3+}(^3P_0\rightarrow)$, and the emission of an ultralow concentration of $Pr^{3+}$ was compensated for by the high concentration of $Tb^{3+}$ [25]. Meanwhile, the intensities of the $Tb^{3+}/Pr^{3+}$ co-doped sample were weaker than those of the single-doped one. In the $Tb^{3+}/Pr^{3+}$:CYA crystal, the adjacent energy positions of $Tb^{3+}:^5D_4$ and $Pr^{3+}:^3P_0$, provide possible paths for energy transfer. As a result of the huge concentration difference between $Tb^{3+}$ (13.87 at.%) and $Pr^{3+}$ (0.38 at.%), the energy transfer process of $Tb^{3+}(^5D_4)\rightarrow Pr^{3+}(^3P_0)$ was more efficient than the backward one, leading to a weaker fluorescence intensity than the single-doped sample.

Based on the following Füchtbauer–Ladenburg (F–L) formula [18], the stimulated emission cross-sections for the $^5D_4\rightarrow^7F_J$ (J = 5, 4) translations can be obtained from polarized fluorescence spectra:

$$\sigma_{em} = \frac{\beta\lambda^5 I(\lambda)}{8\pi cn^2\tau_{rad}\int\lambda I(\lambda)d\lambda} \tag{1}$$

where $\lambda$, $\beta$, $c$, and $I(\lambda)$ refer to the fluorescence wavelength, branching ratio, speed of light, and fluorescence intensity, respectively. The peak emission wavelengths, FWHMs, and emission cross-sections $\sigma^{em}$ of the transitions starting from the $^5D_4$ multiplets of the $Tb^{3+}$:CYA and $Tb^{3+}/Pr^{3+}$:CYA crystals are listed in Table 6. According to Table 6, the FWHMs of the 546 nm emission band in the $Tb^{3+}$:CYA and $Tb^{3+}/Pr^{3+}$:CYA crystals ($\pi$ polarization) were 9.41 nm, 9.31 nm, respectively. The $\pi$ and $\sigma$ polarization emission cross-sections at 546 nm in the green light region were $7.57 \times 10^{-22}$ cm$^2$ and $8.82 \times 10^{-22}$ cm$^2$, respectively, for the $Tb^{3+}$:CYA crystal—slightly larger than the $\pi$ and $\sigma$ polarization emission cross-sections of the $Pr^{3+}/Tb^{3+}$:CYA crystal ($6.99 \times 10^{-22}$ cm$^2$ and $8.55 \times 10^{-22}$ cm$^2$, respectively). These results were also greater than those for $Tb^{3+}$:CaF$_2$ ($5.56 \times 10^{-22}$ cm$^2$) [14] and Ba$_3$TbPO$_4$ ($5.9 \times 10^{-22}$ cm$^2$) [26]. The emission cross-sections of the $^5D_4\rightarrow^7F_4$ transition for $Tb^{3+}$:CYA were calculated to be $0.44 \times 10^{-22}$ cm$^2$ ($\pi$) and $0.34 \times 10^{-22}$ cm$^2$ ($\sigma$), while the results for $Tb^{3+}/Pr^{3+}$:CYA were determined to be $0.35 \times 10^{-22}$ cm$^2$($\pi$) and $0.21 \times 10^{-22}$ cm$^2$ ($\sigma$). The maximum emission cross-section of $Tb^{3+}$:CYA at 587 nm ($0.44 \times 10^{-22}$ cm$^2$) was of the same order of magnitude as that of $Tb^{3+}$:STB crystal ($0.61 \times 10^{-22}$ cm$^2$ *E//Z*) [13].

**Table 6.** Peak emission wavelengths, FWHMs, and emission cross-sections $\sigma^{em}$ of the transitions starting from the $^5D_4$ multiplets of $Tb^{3+}$:CYA and $Pr^{3+}/Tb^{3+}$:CYA crystals.

| Crystal | Transition | Polarization | Peak Wavelength (nm) | FWHM(nm) | $\sigma^{em}$ ($10^{-22}$ cm$^2$) |
|---|---|---|---|---|---|
| | $^5D_4\rightarrow$ | | | | |
| Tb$^{3+}$:CYA | $^7F_5$ | $\pi$ | 546 | 9.41 | 7.57 |
| | | $\sigma$ | 546 | 7.79 | 8.82 |
| | $^7F_4$ | $\pi$ | 587 | 8.43 | 0.44 |
| | | $\sigma$ | 587 | 13.3 | 0.34 |
| Tb$^{3+}$/Pr$^{3+}$:CYA | $^7F_5$ | $\pi$ | 546 | 9.31 | 6.99 |
| | | $\sigma$ | 546 | 6.28 | 8.55 |
| | $^7F_4$ | $\pi$ | 587 | 8.24 | 0.35 |
| | | $\sigma$ | 587 | 10.54 | 0.21 |

In order to explore effects of the doping concentration ratio of $Tb^{3+}$ and $Pr^{3+}$ on the energy transfer process between those two ions, we produced 10at.% $Tb^{3+}$/0.6at.% $Pr^{3+}$, 10at.% $Tb^{3+}$/1at.% $Pr^{3+}$, and 10at.% $Tb^{3+}$/3at.% $Pr^{3+}$ co-doped CYA single-crystal fibers through the micro-pulling-down method. The room-temperature fluorescence spectra in the 530–680 nm range were recorded, as shown in Figure 6. Based on these results, the luminescence intensity of the main bands responsible for $Tb^{3+}$ ions decreased with the increase in the $Pr^{3+}$ ions. This phenomenon can be explained through the differences in the electron shell structures of $Pr^{3+}$ and $Tb^{3+}$. Non-radiative processes were the main energy transfer routes between $Tb^{3+}$ and $Pr^{3+}$. It is therefore assumed that non-radiative energy transfer is carried out with high energy levels from the $Tb^{3+}$ to the $Pr^{3+}$. For the co-doped samples, $Tb^{3+}$ is the dominant luminescence center, as its concentration is as high as 10 at.%. With the increase in the $Pr^{3+}$ concentration, the distance between $Tb^{3+}$ and $Pr^{3+}$ shortened accordingly, and the non-radiative processes between $Tb^{3+}$ and $Pr^{3+}$ intensified, causing a reduction in the luminescence intensity. Similar experimental results were observed in $Tb^{3+}/Pr^{3+}$ co-doped scintillation glass [27]. The large distance between $Tb^{3+}$ and $Pr^{3+}$ might weaken the interaction between them. In the study of Chen et al., energy transfer from $Tb^{3+}\rightarrow Pr^{3+}$, which involved two processes—$Tb^{3+}$:$^5D_4$ + $Pr^{3+}$:$^3H_4\rightarrow Tb^{3+}$:$^7F_6$ + $Pr^{3+}$:$^1I_6$ and $Tb^{3+}$:$^5D_4$ + $Pr^{3+}$:$^3H_4\rightarrow Tb^{3+}$:$^7F_4$ + $Pr^{3+}$:$^3P_0$—was achieved in 0.3 at.% $Tb^{3+}$/0.5 at.% $Pr^{3+}$:CYA phosphor [28]. This result indicates that the dominant energy transfer channel in CYA is $Tb^{3+}\rightarrow Pr^{3+}$, although the two ions are both at low doping levels.

The energy transfer processes between $Tb^{3+}$ and $Pr^{3+}$ are inefficient, and the metal-to-metal intervalence charge transfer (IVCT) processes between d0 electron-configured transition metal ions in oxide crystals and $Pr^{3+}/Tb^{3+}$ have been confirmed to be effective pathways to excite the $Pr^{3+}/Tb^{3+}$ [29]. However, no IVCT process takes place in $Tb^{3+}/Pr^{3+}$:CYA. According to the experimental results of Liu et al., the effective absorption of 5 at.% $Tb^{3+}$:CYA was improved by co-doping with 5 at.% $Pr^{3+}$. Due to the concentration quenching of $Pr^{3+}$, the fluorescence intensity for the main $Tb^{3+}$ emission bands did not decrease, but the corresponding fluorescence lifetime reduced greatly [14]. Thus, in our work, the problem of weak absorption of $Tb^{3+}$ around 487 nm was slightly improved by co-doping with $Pr^{3+}$. However, due to the inefficient energy transfer between $Tb^{3+}$ and $Pr^{3+}$ in compounds with no IVCT processes, the emission of $Tb^{3+}$ in the visible band was slightly weakened by co-doping with $Pr^{3+}$.

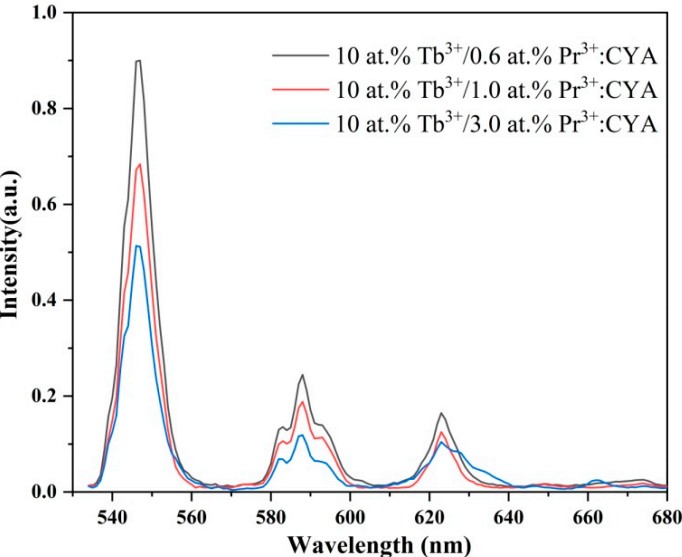

**Figure 6.** The fluorescence spectra of co-doped crystals with different $Pr^{3+}$ concentrations.

### 3.5. Fluorescence Lifetime

Figure 7 shows the fluorescence decay curves of the $^5D_4 \rightarrow {}^7F_5$ transitions in $Tb^{3+}$:CYA and $Tb^{3+}/Pr^{3+}$:CYA crystals excited at 487 nm and 492 nm, respectively. After being fitted, the fluorescence lifetime can be obtained through the following function [30]:

$$I(t) = A_1 e^{\frac{t}{\tau_1}} + A_2 e^{\frac{t}{\tau_2}} + B_1 \tag{2}$$

$$\tau_f = \frac{A_1 \tau_1^2 + A_2 \tau_2^2}{A_1 \tau_1 + A_2 \tau_2} \tag{3}$$

where $I(t)$ refers to the fluorescence intensity as a function of time. The experimental lifetimes $\tau_f$ of the $^5D_4 \rightarrow {}^7F_5$ transitions for the $Tb^{3+}$:CYA and $Tb^{3+}/Pr^{3+}$:CYA crystals were calculated to be 1.43 ms and 1.1 ms, respectively, and the quantum efficiency $\eta$ ($\eta = \frac{\tau_f}{\tau_r}$) was estimated to be 79.2% and 59.14%, respectively. Compared with $Tb^{3+}$:CYA, the shorter fluorescence lifetime of $^5D_4 \rightarrow {}^7F_5$ in the $Tb^{3+}/Pr^{3+}$:CYA crystal may be attributed to the energy transfer process of $Tb^{3+}(^5D_4) \rightarrow Pr^{3+}(^3P_0)$. The energy transfer efficiency from $Tb^{3+}(^5D_4)$ to $Pr^{3+}(^3P_0)$ was calculated to be $\eta = 1 - (1.1/1.43) = 23.07\%$. The energy transfer process decreased the population of $Tb^{3+}$ in the $^5D_4$ state, which had a negative effect on the fluorescence and led to the weakening of the fluorescence lifetime of $Tb^{3+}$. Unfortunately, the energy transfer efficiency value was slightly too large; hence, the impact on the $Tb^{3+}$ fluorescence. Although the emission spectral intensity, emission cross-section, and fluorescence lifetime of $Tb^{3+}$ were decreased through co-doping with $Pr^{3+}$, the absorption cross-section around 487 nm was increased.

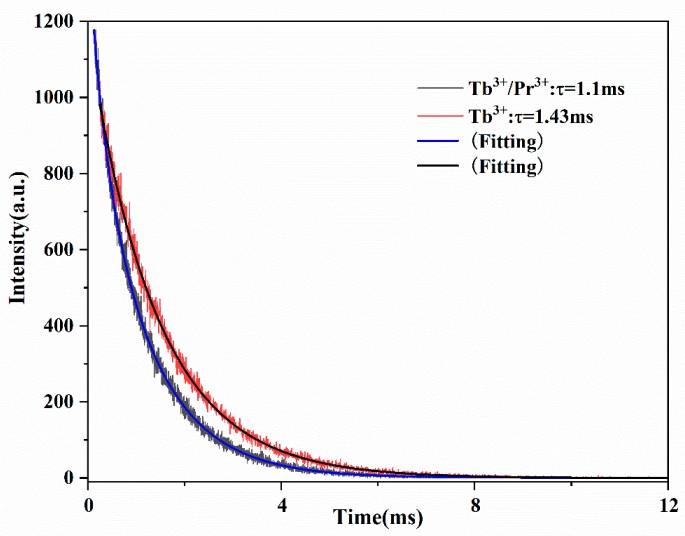

**Figure 7.** Room-temperature decay curve of the $^5D_4$ multiplets for $Tb^{3+}$: CYA and $Tb^{3+}/Pr^{3+}$:CYA.

*3.6. Effects of Temperature on Fluorescence Emission*

Since the laser crystals suffer as a result of high temperatures during long-term operation, the exploration of the thermal stability of the optical properties of the crystals is an important task. Figure 8 shows the relative peak intensity curves of the $Tb^{3+}$:CYA and $Tb^{3+}/Pr^{3+}$:CYA crystals under 487 nm and 492 nm excitation, respectively, with the increase in the temperature from 298 K to 548 K. The relative peak intensity of the two samples decreased almost linearly the increase in temperature. With the increase in the temperature from 298 to 398 K, the intensities of three bright lights at 546 nm (green), 588 nm (yellow), and 623 nm (red) dropped by 24%, 26%, and 27%, respectively, for $Tb^{3+}$:CYA and by 36%, 38%, and 36%, respectively, for $Tb^{3+}/Pr^{3+}$:CYA. Additionally, the chromaticity coordinates of the $Tb^{3+}$:CYA and $Tb^{3+}/Pr^{3+}$:CYA crystals at various temperatures under 487 nm and 492 nm excitation, respectively, are listed in Table 7. The correlated color temperatures (CCTs) were calculated using McCamy's empirical formula [31]:

$$CCT = -449n^3 + 3523n^2 - 6823.8n + 5520.33 \tag{4}$$

$$n = (x - 0.3320)/(y - 0.1858) \tag{5}$$

With the increase in temperature, the chromaticity coordinates of $Tb^{3+}$:CYA varied from (0.370, 0.621) at 298 K to (0.343, 0.636) at 548 K, and the values of $Tb^{3+}/Pr^{3+}$:CYA varied from (0.345, 0.638) at 298 K to (0.246, 0.698) at 548 K; the decrease in the x value and the increase in the y value of the CIE coordinates resulted in all of the coordinates (x,y) invariably being located in the green color region, as shown in Figure 9. This was nothing like the occurrence in the $Tb^{3+}/Pr^{3+}$:Na$_5$Gd(WO$_4$)$_4$ phosphors, in which the most prominent transition was an 648 nm with (0.541, 0.378) coordinates in the orange–yellow region. This was most likely caused by the IVCT processes between $Tb^{3+}$ or $Pr^{3+}$ and transition metal ions (i.e., $Ti^{4+}$, $V^{5+}$, $Nb^{5+}$, $Mo^{6+}$, or $W^{6+}$) with d0 electrons configured in oxide crystals [32]. The results indicated that the $Tb^{3+}$:CYA and $Tb^{3+}/Pr^{3+}$:CYA crystals possessed good thermal stability of their optical properties, as well as potential for green laser applications with a wide temperature range.

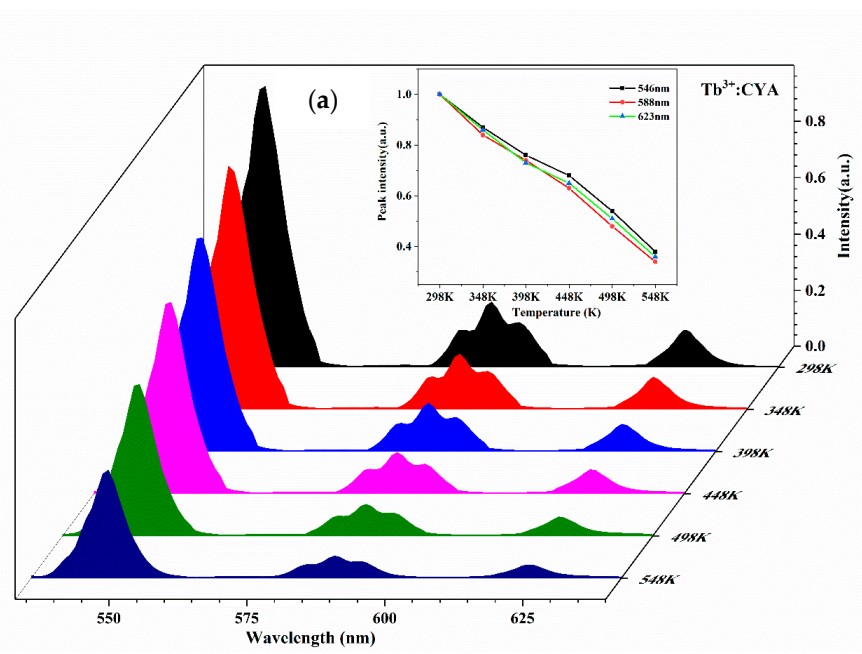

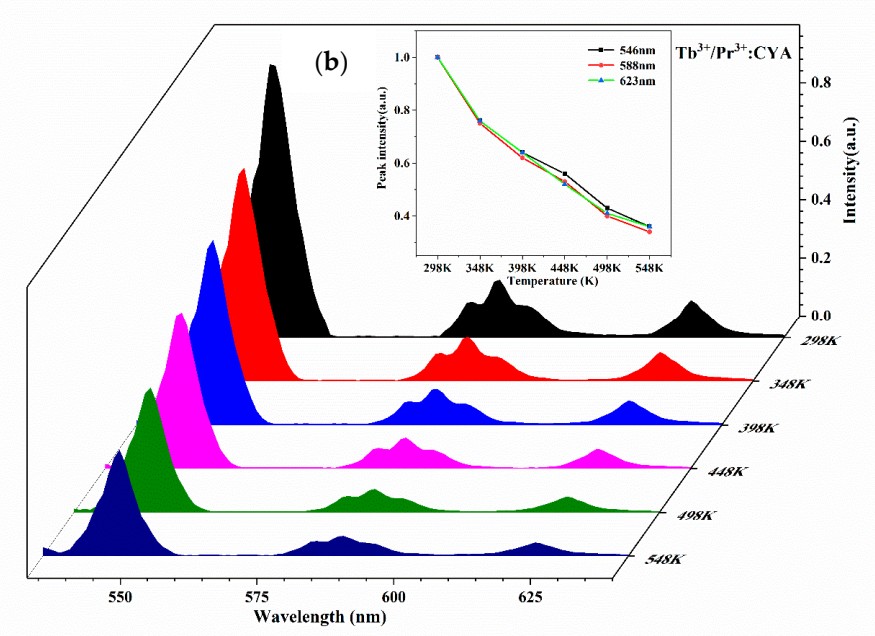

**Figure 8.** Temperature dependence of the fluorescence spectra of (**a**) Tb³⁺:CYA and (**b**) Tb³⁺/Pr³⁺:CYA at 546 nm, 588 nm, and 623 nm.

**Table 7.** The chromaticity coordinates of the Tb³⁺:CYA and Tb³⁺/Pr³⁺:CYA crystals at various temperatures.

| Temperature (K) | Tb³⁺:CYA (CIE) | | | Tb³⁺/Pr³⁺:CYA (CIE) | | |
|:---:|:---:|:---:|:---:|:---:|:---:|:---:|
| | X | Y | CCT (K) | X | Y | CCT (K) |
| 298 K | 0.370 | 0.621 | 4951 | 0.345 | 0.638 | 5327 |
| 348 K | 0.367 | 0.622 | 4995 | 0.313 | 0.658 | 5800 |
| 398 K | 0.366 | 0.623 | 5010 | 0.289 | 0.673 | 6150 |
| 448 K | 0.363 | 0.625 | 5056 | 0.282 | 0.678 | 6250 |
| 498 K | 0.358 | 0.627 | 5130 | 0.268 | 0.687 | 6450 |
| 548 K | 0.343 | 0.636 | 5355 | 0.246 | 0.698 | 6767 |

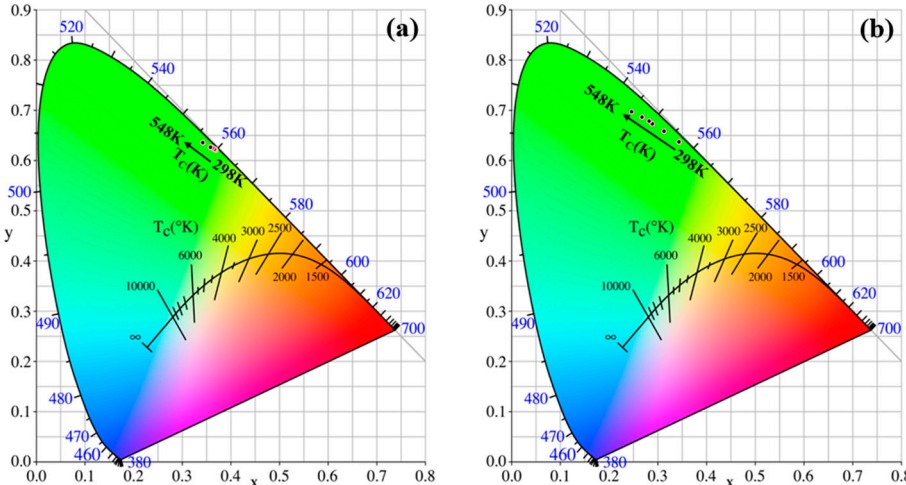

**Figure 9.** The CIE 1931 chromaticity diagrams of (**a**) $Tb^{3+}$:CYA ($\lambda_{ex}$ = 487 nm) and (**b**) $Tb^{3+}/Pr^{3+}$:CYA ($\lambda_{ex}$ = 492 nm) crystals at various temperatures.

## 4. Conclusions

Single crystals of 13.87 at.% $Tb^{3+}$ single-doped and 13.71 at.% $Tb^{3+}$/0.38 at.% $Pr^{3+}$ co-doped CYA were produced by the Czochralski method. The polarized spectra and fluorescence decay curves were studied in detail. Through the incorporation of $Pr^{3+}$, the absorption cross-section around 487 nm was increased from $1.53 \times 10^{-22}$ cm² to $5.53 \times 10^{-22}$ cm² for the $\pi$ polarization. The J–O intensity parameters $\Omega_t$ (2, 4, 6), fluorescence branch ratios ($\beta$), and radiation lifetimes ($\tau_{rad}$) were calculated. For the $Tb^{3+}$:CYA crystal, the stimulated emission cross-sections of the $^5D_4 \rightarrow {}^7F_5$ and $^7F_4$ transitions were calculated to be $7.57 \times 10^{-22}$ cm² and $0.44 \times 10^{-22}$ cm² for $\pi$ polarization, respectively, which were larger than the values for the $Tb^{3+}/Pr^{3+}$:CYA crystal. The fluorescence lifetime of the $^5D_4$ level was measured to be 1.41 ms and 1.1 ms with quantum efficiency of 79.2% and 59.14% for $Tb^{3+}$:CYA and $Tb^{3+}/Pr^{3+}$:CYA, respectively. All of the results show that $Tb^{3+}$:CYA and $Tb^{3+}/Pr^{3+}$:CYA crystals may be potential media for the operation of visible-range lasers. However, $Pr^{3+}$ may not be a good candidate for use as a sensitizing ion for $Tb^{3+}$ to strengthen the visible emission in CYA crystals.

**Author Contributions:** Y.W., J.C., and Y.S. conceived and designed the experiments. J.C. and Z.W. carried out the experiments. J.C., Z.W., J.H. and Y.G. analyzed the data and discussed the results. Y.W. and J.C. wrote the paper. Y.W., Y.S., C.T., and Y.Y. reviewed the paper. All authors have read and agreed to the published version of the manuscript.

**Funding:** This work is has been supported by the National Natural Science Foundation of China (grant 11764014, 61765002, 61905099, 12104194), the Natural Science Foundation of Jiangxi Province (No.20202ACBL202003, 20202ACBL214020), and Jiangxi Provincial Key Laboratory of Functional Molecular Materials Chemistry (20212BCD42018).

**Data Availability Statement:** The data used in this study are available upon request.

**Conflicts of Interest:** The authors declare no conflict of interest.

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
