# Peer review of "Study of the Optical Features of Tb3+:CaYAlO4 and Tb3+/Pr3+:CaYAlO4 Crystals for Visible Laser Applications"

_crystals, doi:10.3390/cryst12121729_

Round 1

Reviewer 1 Report

In this paper, single crystals Tb3+ and Tb3+/Pr3+ doped CaYAlO4 were prepared and characterized at room temperature. The manuscript is well organized and written and the results have clearly shown that the prepared crystal is a good candidate for visible laser for a wide temperature range. I have the following minor comments, otherwise, I suggest publishing the paper:  

 -       Material and Method: The methods of sample preparation should be explained in more detail, not only referring to references.   

-       In Conclusion, line 301: I suggest rewording the first sentence, preferably, not starting with numbers.

-       The quality of figures must be high enough to be able to see clear data when zooming in. 

Reviewer 2 Report

The paper “Optical features study of Tb3+:CaYAlO4 and Tb3+/Pr3+:CaYAlO4 crystals for visible laser application” presents growing and characterization of Tb3+:CaYAlO4 and Tb3+/Pr3+:CaYAlO4 single crystals. Authors conclude that Tb3+:CYA crystals can be used as active laser media in visible region, but Pr is a bad co-dopant for Tb.

I recommend accepting this paper after minor revision.

1. Crystals grown by Czochralski are quite sensitive to many technological factors during growth. Thus I insist that authors provide information on which plant the crystals were grown in the form MODEL (manufacturer, Country of manufacturer, City) in the Methods section of the manuscript.

2. It is unclear why authors have chosen exactly these methods (polarized absorption, etc.). It would be great if authors write in more detail which exactly properties of grown crystal can be evaluated by these methods.

3. All used chemicals should be provided in the following form:

CaCO3 (Purity (99.9 OR 67.254 OR not more than 10^-3 of 23 impurities), Manufacturer, Country of manufacturer)

All mentioned equipment should be provided in the following form:

ICP-MS MODEL (manufacturer, Country of manufacturer, City)

4. Inserts on Figure 8 are too small. Please provide bigger figures. The numbers on the temperature scale should not be slanting, so it is difficult to distinguish them.

5. I suggest the authors to pay more attention to the English language of the manuscript. It is required to correct wrong forms of the verb, inconsistency of tenses, extra or missing spaces.

6. Journal titles should be abbreviated in accordance with generally accepted norms. Some journals are abbreviated incorrectly and some are not abbreviated at all. I also recommend adding DOI to all publications that have this identifier.
